# IGF-1R/mTOR Targeted Therapy for Ewing Sarcoma: A Meta-Analysis of Five IGF-1R-Related Trials Matched to Proteomic and Radiologic Predictive Biomarkers

**DOI:** 10.3390/cancers12071768

**Published:** 2020-07-02

**Authors:** Hesham M. Amin, Ajaykumar C. Morani, Najat C. Daw, Salah-Eddine Lamhamedi-Cherradi, Vivek Subbiah, Brian A. Menegaz, Deeksha Vishwamitra, Ghazaleh Eskandari, Bhawana George, Robert S. Benjamin, Shreyaskumar Patel, Juhee Song, Alexander J. Lazar, Wei-Lien Wang, Razelle Kurzrock, Alberto Pappo, Peter M. Anderson, Gary K. Schwartz, Dejka Araujo, Branko Cuglievan, Ravin Ratan, David McCall, Sana Mohiuddin, John A. Livingston, Eric R. Molina, Aung Naing, Joseph A. Ludwig

**Affiliations:** 1Department of Hematopathology, the University of Texas MD Anderson Cancer Center, Houston, TX 77030, USA; hamin@mdanderson.org (H.M.A.); deeksha1211@gmail.com (D.V.); geskandari@houstonmethodist.org (G.E.); bhawanantu@gmail.com (B.G.); 2Department of Nuclear Medicine, the University of Texas MD Anderson Cancer Center, Houston, TX 77030, USA; amorani@mdanderson.org; 3Department of Pediatrics, the University of Texas MD Anderson Cancer Center, Houston, TX 77030, USA; ndaw@mdanderson.org (N.C.D.); cuglievan.branko@gmail.com (B.C.); DMcCall1@mdanderson.org (D.M.); SMohiuddin@mdanderson.org (S.M.); 4Department of Sarcoma Medical Oncology, the University of Texas MD Anderson Cancer Center, Houston, TX 77030, USA; slamhamedi@mdanderson.org (S.-E.L.-C.); rbenjami@mdanderson.org (R.S.B.); spatel@mdanderson.org (S.P.); daraujo@mdanderson.org (D.A.); RRatan@mdanderson.org (R.R.); JALivingston@mdanderson.org (J.A.L.); 5Department of Investigational Cancer Therapeutics, 7Department of Biostatistics, The University of Texas MD Anderson Cancer Center, Houston, TX 77030, USA; vsubbiah@mdanderson.org (V.S.); anaing@mdanderson.org (A.N.); 6Baylor College of Medicine, Department of Surgery, Breast Surgical Oncology, Houston, TX 77030, USA; Menegaz@bcm.edu (B.A.M.); eric.molina@bcm.edu (E.R.M.); 7Department of Biostatistics, the University of Texas MD Anderson Cancer Center, Houston, TX 77030, USA; JSong1@mdanderson.org; 8Department of Pathology, the University of Texas MD Anderson Cancer Center, Houston, TX 77030, USA; alazar@mdanderson.org (A.J.L.); wlwang@mdanderson.org (W.-L.W.); 9Center for Personalized Cancer Therapy, University of California San Diego (UCSD) Moores Cancer Center, San Diego, CA 92037, USA; rkurzrock@ucsd.edu; 10Department of Pathology, St. Jude’s Cancer Research Hospital, Memphis, TN 38105, USA; alberto.pappo@stjude.org; 11Department of Pediatric Oncology, Cleveland Clinic, OH 44195, USA; andersp@ccf.org; 12Division of Hematology & Oncology, Columbia University Medical Center, New York, NY 10032, USA; gks2123@cumc.columbia.edu

**Keywords:** Ewing sarcoma, biomarker, pIGF-1R, PET/CT, drug response, biological therapies

## Abstract

*Background *: Ten to fourteen percent of Ewing sarcoma (ES) study participants treated nationwide with IGF-1 receptor (IGF-1R)-targeted antibodies achieved tumor regression. Despite this success, low response rates and short response durations (approximately 7-weeks) have slowed the development of this therapy. *Methods*: We performed a meta-analysis of five phase-1b/2 ES-oriented trials that evaluated the anticancer activity of IGF-1R antibodies +/− mTOR inhibitors (mTORi). Our meta-analysis provided a head-to-head comparison of the clinical benefits of IGF-1R antibodies vs. the IGF-1R/mTOR-targeted combination. Available pretreatment clinical samples were semi-quantitatively scored using immunohistochemistry to detect proteins in the IGF-1R/PI3K/AKT/mTOR pathway linked to clinical response. Early PET/CT imaging, obtained within the first 2 weeks (median 10 days), were examined to determine if reduced FDG avidity was predictive of progression-free survival (PFS). *Results*: Among 56 ES patients treated at MD Anderson Cancer Center (MDACC) with IGF-1R antibodies, our analysis revealed a significant ~two-fold improvement in PFS that favored a combination of IGF-1R/mTORi therapy (1.6 vs. 3.3-months, *p* = 0.042). Low pIGF-1R in the pretreatment specimens was associated with treatment response. Reduced total-lesion glycolysis more accurately predicted the IGF-1R response than other previously reported radiological biomarkers. *Conclusion*: Synergistic drug combinations, and newly identified proteomic or radiological biomarkers of IGF-1R response, may be incorporated into future IGF-1R-related trials to improve the response rate in ES patients.

## 1. Introduction

Ewing sarcoma (ES), the second most common bone cancer in adolescents and young adults (AYA), is caused by a pathognomonic genomic translocation that juxtaposes an N-terminal *EWSR1* gene with one of several E26 transformation-specific (ETS) genes (typically *FLI1 or ERG*) [1]. The resulting fusion protein can serve as a pioneer factor that alters the transcription of more than 500 genes and, thereby, perturbs pathways critical for oncogenesis and metastasis [2,3,4]. Among those, the IGF-1 receptor (IGF-1R) pathway has been extensively studied given its essential role in promoting ES transformation from primitive neuroendocrine or mesenchymal stem cells (MSCs) [5,6,7,8].

IGF-1R-targeted monoclonal antibodies (mAbs) were first tested as a treatment for ES more than ten years ago [9,10]. Clinical results since then have been mixed, as summarized by Fleuren et al.; ~10–14% of ES patients had significant (albeit short-lived) tumor shrinkage, while most others had rapid tumor progression [11]. The low response rate and short duration of activity among ES patients, together with the limited effectiveness in epithelial malignancies, severely curtailed the enthusiasm of IGF-1R Abs as a potential antineoplastic agent. Single-agent IGF-1R mAbs are no longer being explored as an experimental treatment option for ES patients and the most recent Children’s Oncology Group’s study, AEWS1221, was terminated in March 2019 after ganitumab (Amgen’s IGF-1R-targeted Ab; AMG479) failed to synergize with standard-of-care cytotoxic chemotherapy in newly diagnosed metastatic ES patients. An increased incidence of pneumonitis was also noted in the experimental group in patients that had previously received adjuvant whole-lung radiation.

Though a development path for single-agent IGF-1R mAbs has ceased, these agents were well tolerated and remarkably active in a subset of patients. Supporting the latter point, as compared to more than twenty other experimental therapies tested in ES patients over the last decade at MD Anderson Cancer Center (MDACC), Subbiah et al. noted that only IGF-1R mAbs +/− mTOR inhibitors (mTORi) effectively suppressed tumor growth [12,13]. With this signal of activity, our laboratory and others worldwide sought to enhance the effectiveness of IGF-1R targeted therapies by identifying synergistic drug combinations and determining biomarkers that predict which ES patients are most likely to respond.

The first suggestion that IGF-1R mAb activity might be enhanced when co-administered with other agents came from ES cell lines and xenografts that adapted to IGF-1R mAbs and IGF-1R-targeted small molecules by upregulating IR-α [14,15], IRS-1 [16], Stat3 [16], MSTR1 [17] and other proteins throughout IGF-1R/PI3K/mTOR signaling cascade. Around the same time, derepression of IRS-1—immediately downstream of IGF-1R—was described as a potent mechanism used by cancer cells to quickly evade single-agent mTORi [18,19,20,21]. With converging evidence that IGF-1R mAbs counteracted this IRS-1 activation, IGF-1R/mTORi combinations were tested extensively in the preclinical and clinical settings [22,23,24]. A phase-2 trial of cixutumumab/temsirolimus, first published by Naing et al. in 2012, demonstrated nearly a 29% response rate and 7-fold longer median response duration (>14 months) in ES patients [25]. Those results were significantly better than response rates observed when IGF-1R mAbs or mTORi were used individually in ES or other diverse sarcomas [26,27]. Two subsequent trials were unable to replicate that promising result, and, in both cases, the mTORi was either dose reduced in a significant number of patients or initiated at a subtherapeutic dose tolerable in children [28,29]. Since each of those trials were single-arm studies, no prospective or retrospective head-to-head comparison of IGF-1R/mTOR vs. IGF-1R-directed therapy has ever been conducted.

In the present study, we analyzed the clinical outcome of 56 advanced-stage ES patients treated at MDACC from 2007–2013 on one of five different nationwide IGF-1R-based clinical trials (three single-agent IGF-1R mAbs; two IGF-1R/mTOR inhibitors). Among this similar set of patients treated at a single institution by the same team of ES specialists, our retrospective analysis indicates a statistically significant benefit obtained with the inclusion of mTORi to IGF-1R Abs. Though this finding highlights the need for additional IGF-1R/mTORi-based studies, their design should maintain a therapeutic mTORi dose and, ideally, better select for IGF-1R/mTORi responders using response biomarkers. Already, a better understanding of the pathobiology of mTORi-associated mucositis suggests that topical or oral steroids can suppress the pronounced inflammatory lesions that would otherwise lead to mTORi dose reductions [30]. Since IGF-1R response biomarkers have remained elusive, we measured archival tissue specimens from all available IGF-1R mAb-treated patients to determine if pIGF-1R, pAkt, or other immunohistochemical protein markers correlated with clinical outcomes. Furthermore, as an early response biomarker, we sought to validate key radiologic findings from the SARC11 study reported by Koshkin et al., which early changes to FDG avidity predict OS [31].

## 2. Results

### 2.1. Joint Suppression of IGF-1R/mTOR is Superior to Single-Agent IGF-1R mAbs

Despite preclinical evidence that IGF-1R mAbs can circumvent mTORi-induced IRS-1 activity, IGF-1R/mTORi combinations have not been retrospectively or prospectively compared to single-agent IGF-1R Abs either. Using a clinical protocol approved by our Institutional Review Board (IRB), we performed a single-institution meta-analysis to assess the clinical outcome of all ES patients who had received an IGF-1R Ab alone or in combination with temsirolimus. Except for one patient who received cixutumumab/temsirolimus elsewhere, all patients were treated at MDACC by a limited number of investigators that specialize in ES. This uniform referral pattern and consistent institutional treatment approach afforded an unbiased comparison of the different IGF-1R based therapies.

Five ES-specific IGF-1R-related clinical trials were available for analysis (Figure 1A), including two that evaluated single-agent teprotumumab (better known as R1507), one that assessed single-agent robatumumab, and two that used the cixutumumab/temsirolimus combination. With one exception, all trials were phase 2. Kurzrock et al. included a phase-1 lead-in that dosed patients at either 1 mg/kg or 3 mg/kg weekly before subsequently settling upon a recommended phase 2 dose of 9 mg/kg in an expansion cohort; two of the patients treated at lower R157 doses were later dose escalated to 9 mg/kg after completing their cycle-1 pharmacokinetic studies. Table 1 lists the principal investigators of each nationwide study, the MDACC IRB numbers, and select demographic information. Note, the enrollment number (*n*), progression-free survival (PFS), and OS are restricted to the MDACC-treated subset of patients for which complete clinical data existed and may not reflect the broader nationwide dataset. Full trial details may be found in the referenced original publications. Though the eligibility requirements varied slightly, each trial enrolled advanced-stage ES patients that had locally recurrent or multifocal chemoresistant disease that failed multiple prior chemotherapy regimens.

Our analysis combined both R1507 trials into a single group (n = 22; Figure 1B). Subsequently, to improve the statistical power of our analysis, the combined R1507 and robatumumab group (*n* = 40) was compared to the cixutumumab/temsirolimus combination (*n* = 21), resulting in a 2:1 ratio of single vs. dual therapy. Because five of 56 unique patients (8.9%) initially received single-agent IGF-1R Abs and later enrolled onto the cixutumumab/temsirolimus trial after progressing on R1507 or robatumumab, there were 61 treatment records for 56 patients. As discussed in the Methods section, the most conservative statistical approach excluded those five patients from analysis, leaving 51 patients that had been treated exclusively with an IGF-1R Ab (*n* = 35) or IGF-1R/mTORi combination (*n* = 16) (Figure 1C). No differences in age (median 20.9 vs. 19.85 years; *p*-value = 0.7072), gender, or other patient characteristics were observed between patients who received a single IGF-1R Ab vs. the combination. Neither IGF-1R Ab was superior to the other with respect to response rates (R1507: 27.3% vs. Robatumumab: 33.3%; *p*-value 0.6773).

Contrary to our expectation, a higher proportion of patients achieved a partial response when treated using single-agent IGF-1R Abs (31.4% vs. 18.8%, *p* = 0.0003; Figure 1D). However, the clinical benefit ratio (i.e., stable disease (SD), partial response (PR), and complete response (CR)) strongly favored the combination therapy since significantly more patients achieved SD in the combination group (75.0% vs. 37.1%, *p* = 0.0121; Figure 1E). Alternative statistical models that included patients who had initially received an IGF-1R mAb, and later an IGF-1R/mTORi combination, reached the same conclusions (Appendix A). Given the aggressive nature of ES, the clinical response to single-agent IGF-1R mAbs was nearly dichotomous. Most non-responders progressed during their first 6-week restaging visit, whereas the PFS among responders varied widely (Figure 1F). As shown in Figure 1G, improved clinical benefit among the IGF-1R/mTOR-treated group translated into a statistically significant increase in PFS (*p* = 0.042); median PFS of the IGF-1R and IGF-1R/mTOR-treated groups was 1.6 months and 3.3 months, respectively, and the hazard ratio (HR) favored combination therapy (HR = 1.95, 95% CI, 1.023–3.719). A non-statistical trend toward improved OS was observed for the combination therapy group; however, the meta-analysis was not sufficiently powered to detect a survival difference (Figure 1H).

### 2.2. Early PET/CT Imaging Predicts Clinical Response at 6 Weeks and Progression-Free Survival (PFS)

The usefulness of early ^18^F-FDG PET imaging—obtained just 8 days (range 8–14 days) after beginning R1507 IGF-1R mAb treatment—has been shown by Sarcoma Alliance for Research and Collaboration (SARC) investigators to predict a 6-week response, PFS, and OS. The first of two SARC publications based upon their 115-patient study, by Hyun et al., used PERCIST 1.0 criteria [34], whereas the second, by Koskin et al., assessed the value of 2-dimensional imaging (e.g., WHO and RECIST criteria) and 3-dimensional volumetric imaging [31]. Among their key findings, PERCIST was more sensitive than RECIST at detecting an early clinical response and associated with a 35% response rate, which they posed in retrospect, might have been worthy of regulatory approval had this metric been used rather than WHO. To determine if the SARC research findings held when validated against other IGF-1R-based ES trials, including those reliant upon other IGF-1R mAbs (e.g., robatumumab), we identified ES patients that had enrolled in one of five IGF-1R-related clinical trials at MDACC. The patients analyzed included those who received at least one dose of IGF-1R-targeted therapy and underwent an early ‘research-only’ PET/CT imaging at a median of 10-days after the first dose (range 6–26 days), much sooner than the 6-week time points used to assess treatment response in ES (Appendix A).

Thirteen of the thirty-six patients that met that criteria overlapped with the previously reported SARC-sponsored study, and the remaining patients had enrolled in one of the four other trials, which provided single-agent IGF-1R Abs (*n* = 19) or the IGF-1R/mTOR inhibitor combination (*n* = 4). Early PET/CTs, obtained at prespecified time points 8- and 13-days after the first IGF-1R mAbs dose were, respectively, integrated into the trials conducted by Pappo et al. (2011) and Anderson et al. (2016). Though not mandated by the other studies, MDACC trial participants optionally underwent early PET/CT imaging at ~1-week to closely match the time point stipulated by the SARC11 study. Four metrics of tumor metabolic activity were assessed: SUV_Max_ (EORTC criterion), SUL_Max_, SUL_Peak_ (PERCIST criterion), and total lesion glycolysis (TLG). As reported elsewhere, the PERCIST 1.0 response was based on the highest mean standard uptake values (SUV), corrected for lean body mass, within a 1-cm^3^ sphere from the most FDG-avid part of the tumor (SUL_Peak_). Two additional anatomical, tumor measurement-based metrics (WHO and RECIST 1.1 criteria) were obtained from the concordant CT of the PET/CT and/or accompanying chest/body CT by an experienced radiologist at baseline (PET/CT_Baseline_) and within the first cycle of IGF-1R-targeted therapy (PET/CT_Early_).

All six image-based metrics were associated with tumor response, clinical benefit, PFS, and OS (*p*-values listed in Appendix A). Of those parameters, TLG demonstrated the highest performance in predicting tumor response and clinical benefit, with receiver operating characteristics (ROCs) of 0.92 and 0.88, respectively (Table 2). Interestingly, because ES is such an aggressive tumor, the mere absence of tumor growth in the first two weeks of treatment, as assessed by WHO and RECIST criteria, placed patients into a favorable response category (Figure 2A,B). Reduced FDG avidity by standard metrics (e.g., EORTC, PERCIST) and TLG slightly outperformed size-based measures in predicting each patient’s best clinical response (Figure 2C–E). Though early research images were infrequently performed in the subset of patients that received cixutumumab/temsirolimus, they also seemed to be predictive of the best clinical response (Appendix A). The density plots shown in Figure 2 indicate the proportion of patients that had a partial response, stable disease, or progression as their best clinical response for each size (Figure 2A,B) or metabolic (Figure 2C–E) PET/CT_Early_ imaging parameter.

Among the subgroup that achieved clinical benefit (CR/PR/SD) by RECIST criteria, early research imaging could not delineate patients that achieved a complete/partial response from those with disease stabilization; TLG came close but failed to reach statistical significance (Appendix A). To determine if PET/CT_Early_ CT- and PET-based parameters enhanced predictive accuracy when used together, a multivariable logistic regression model was constructed using WHO- or RECIST-based measures of tumor size together with each of the three PET parameters. The RECIST/TLG combined response metric yielded the best predictive capacity, with an ROC of 0.93 in predicting tumor response (Appendix A). This combination of metabolic and anatomical data was, however, only marginally better than the results achieved using CT or PET parameters by themselves.

### 2.3. Low pIGF-1R Predicts 6-Weeks Response to IGF-1R Targeted Abs

To extend our analysis of early response biomarkers, we analyzed archival tissue specimens from IGF-1R mAb-treated patients. Consistent with prior publications, total IGF-1R expression did not predict outcome [28,35]. However, as part of our ‘*n* = 1 program’ that comprehensively evaluates the genomic and proteomic profile of unusual responders, pIGF-1R expression was absent or barely detectable in stained tissue sections.

We subsequently launched a confirmatory analysis of the entire population of ES patients that had received IGF-1R-targeted therapies at our institution, blinding the reviewing pathologist to the treatment outcome. Of fifty-six patients treated with IGF-1R mAbs +/− temsirolimus, pretreatment formalin-fixed paraffin-embedded (FFPE) tissue blocks were available for eighteen patients. Immunohistochemistry of tissue sections from those patients was examined by an experienced pathologist for expression of pIGF-1R and total IGF-1R (Figure 3A), in addition to other proteins within the IGF-1R/mTOR signaling cascade, using a 4-level scale (e.g., absent, 1+, 2+, and 3+).

Contrary to the pattern we hypothesized, low pIGF-1R expression was associated with a better response to therapy. All six patients with pIGF-1R-negative tumors had a partial response, whereas only 33% of the pIGF-1R positive tumors responded (Table 3); individual response data is provided in Appendix A. Notably, the absence of detectable pretreatment pIGF-1R was predictive of RECIST response (*p* = 0.0128) and clinical benefit (*p* = 0.0377). The positive predictive value (PPV) of pIGF-1R-negative expression was 100% (95% CI 54–100%) and negative predictive value (NPV) 67% (95% CI 35–90%).

Next, we assessed whether a patient’s pretreatment pIGF-1R status was associated with their PET/CT_Early_ metrics (Figure 3B–F), the assumption being that predicted responders—based upon their pIGF-1R-negative status—would also exhibit favorable response metrics in their day 9–14 early PET/CTs. Shown in Figure 3B–F for every early radiological metrics of response (RECIST, WHO, PERCIST), tumors that lacked pIGF-1R were associated with better radiological response. This association was most evident in the PET-based PERCIST, EORTC, and TLG metrics described earlier that had best predicted the week-6 response and PFS.

## 3. Discussion

Three significant insights emerge from our present work. First, our meta-analysis demonstrated that the dual inhibition of IGF-1R/mTOR with the cixutumumab/temsirolimus combination improved PFS compared to single-agent IGF-1R-directed treatment with teprotumumab (R1507) or robatumumab. Second, we identified a novel protein biomarker pIGF-1R from pretreatment archival tumor specimens that appears to be predictive of ES tumor response to IGF-1R Ab-based therapy. Third, we independently validated that early functional imaging, acquired at a median of 10 days after therapy initiation, is a clinically useful predictor of clinical response, PFS, and OS.

Putting these findings into context, our meta-analysis is the first to directly compare joint IGF-1R/mTOR-targeting head-to-head against single-agent IGF-1R Abs in patients with ES. Though different drug combinations co-targeting IGF-1R and mTOR have been reported in preclinical studies, the cixutumumab/temsirolimus drug combination was the only one advanced to patients at MDACC and available for analysis. Nevertheless, it sets the stage for drug optimization in future trials that co-target two or more proteins within the IGF-1/IGF-1R/PI3K/mTOR signaling cascade. Among the more than half-dozen IGF-1R mAbs initially studied as single-agent therapy for ES, only ganitumab remains in clinical development [11]. The clinical development of linsitinib (OSI-906) and other small-molecule tyrosine kinase inhibitors (TKIs) that competitively block the ATP-binding cleft of IGF-1R and insulin receptor alpha (IR-α) has been abandoned due to dose-limited hyperglycemia, which resulted from undesirable binding to the IR-β isoform responsible for insulin-mediated metabolic signaling [36]. Neutralizing Abs against the IGF ligands themselves were well-tolerated but had inadequate clinical activity [37]. Moreover, of the two ligand-targeted drugs tested in early-phase clinical trials, only xentuzumab (BI 836845) persists following the halt of MEDI-573 development with Medimmune’s acquisition by AstraZeneca [36].

Taken together, though notable RECIST responses were observed in 10–14% of the ES patients treated with single-agent IGF-1R-targeted therapies, the most likely development path that remains is one that capitalizes upon highly synergistic, rationally selected drug combinations that maximally ablate the entire signaling cascade upstream and downstream of mTOR. Naturally, mTORi take center stage in this, but their derepression of IRS-1 and secondary up-regulation of Akt severely curtails their activity when used alone [18,19,20,21]. Notably, since the COG and MSKCC studies failed to replicate the initial positive data reported by Naing et al. (2012), possibly due to mTORi dose reductions in response to elevated transaminases, oral mucositis, or hyperglycemia. Future trials will need to better alleviate those side effects to prevent mTORi dose reductions. Discussions at AACR’s 2018 annual meeting suggested that such dose reductions were probably unwarranted and might have been addressed differently using aggressive supportive care (personal communication); oral glutamine solutions, for example, can reduce mucositis and mild to moderate hyperglycemia can be effectively managed using metformin. Relevant to our meta-analysis and the use of mTORi with IGF-1R Abs, we excluded five of fifty-six patients treated at our institution that had enrolled on the cixutumumab/temsirolimus study after their tumors had progressed on a previous IGF-1R Ab study given our concern that prior IGF-1R Ab exposure would bias the later trial’s response rates. Of those five who had enrolled in both studies, the only patient that had a second clinical response was one that previously responded to single-agent IGF-1R Abs; thus, mTORi does not appear to benefit those whose tumors are IGF-1R Ab insensitive [25].

Although effective drug combinations are vital, trial enrichment will rely upon predictive biomarkers. As noted by Haluska et al., “The challenge in unlocking the potential for IGF-1R targeted therapies lies in the need of predictive biomarkers” [38]. Trial enrichment for IGF-1R responders using protein or radiologic predictive biomarkers may allow trials to achieve the same statistical significance with fewer patient numbers, and this would theoretically allow more early-phase clinical trials to be conducted, even in rare tumors like ES. While the paradigm for biomarker identification and usage is straightforward, no single protein biomarker has emerged until now. Past phase 2 studies by-and-large did not include pretreatment correlative studies or tumor biopsies that would have been helpful in biomarker development. An exception was the study conducted by Schwartz et al. (2013), which assessed pre- and post-treatment tumor expression levels of IGF-1R, pAkt, and pS6 from patients with diverse sarcoma subtypes treated with cixutumumab/temsirolimus. Interestingly, though none of the IHC parameters correlated with 12-week PFS in that study, a post-hoc analysis of the ES subset revealed a statistically significant three-fold improvement in median PFS in the ES patients who were IGF-1R negative [28]. This immunohistochemical finding fits with our finding that pIGF-1R-negative patients have a better clinical response, PFS, and OS.

In contrast to our expectations, we were surprised to discover that the absence of pIGF-1R in the nucleus or cytoplasm predicted the response to IGF-1R Abs. As future mechanistic studies explore why low or absent pIGF-1R expression was associated with a higher response rate, it is essential to note that our IHC analysis used a mAb against the pIGF-1R tyrosine 1161 residue because it outperformed other phospho-targeted Abs (ones targeting tyrosine 950 and 1131/1135/1136) (Appendix A). As the biological impact of this phosphorylation site is less well characterized, future studies should determine whether pIGF-1R(Y1161) is substantially similar to the other phosphorylation sites in its effect upon downstream canonical and non-canonical signaling. The IGF-1R phosphorylation state of sarcoma patients treated with IGF-1R Abs has not been reported in other studies, nor has pIGF-1R been linked to prognosis more generally. Intriguingly, the French research team led by Asmane et al. reported in 2012 that exclusive nuclear staining of *total* IGF-1R was a possible biomarker of IGF-1R Ab response among a group of 16 sarcoma patients that included diverse sarcoma subtypes, including osteosarcoma (*n* = 4), liposarcoma (*n* = 6), rhabdomyosarcoma (*n* = 1), synovial sarcoma (*n* = 1), desmoplastic small round cell sarcoma (*n* = 1), and three patients with ES [39]. Though our analysis also evaluated total IGF-1R, its expression had no relationship with tumor response, PFS, or OS.

Complementing the discovery of a novel protein biomarker, we have validated the utility of early PET/CT imaging as a radiologic biomarker predictive of IGF-1R mAb response in ES while extending those results to include TLG, a volumetric metabolic parameter of the tumor burden derived using semi-automated segmentation software from routine PET imaging. TLG is calculated by multiplying the metabolic volume by the SUV mean of the entire tumor. As it represents both the hypermetabolic activity of the tumor as well as the tumor size, it appears to be more accurate than other PET metrics [40]. TLG is an optional tool used to assess total tumor burden but is not currently included within formal criteria to measure response by EORTC or PERCIST. 

Having validated the use of PET/CT as a very early indicator of treatment response in ES, we favor using this imaging modality jointly with pIGF-1R testing in any subsequent IGF-1R-related study to predict clinical response, PFS, and OS. Though pIGF-1R is a true pretreatment biomarker, its reliance upon semi-quantitative IHC is subject to the interpretation of experienced pathologists and will require CLIA-certification before it can be used as an enrollment criterion. In contrast, while a week-long IGF-1R Ab test-dose would need to be given, functional imaging criteria using PERCIST or TLG have clear quality-control guidelines that facilitate reproducible interpretations throughout the world. In choosing the precise PET/CT_Early_ metrics cutoff used to classify ES patients as predicted responders, Hyun et al. used a dichotomized percent change in SUL_Peak_—split at 10.5%—to estimate OS [34]. Since no PET/metabolic metric classifier perfectly separated responders from non-responders, the density plots in Figure 2 allow investigators to choose a customized PET response criterion that optimally enriches for responders while excluding most non-responders.

We recognize potential limitations in our study. First, we evaluated only those ES patients who enrolled in clinical trials at MD Anderson. Second, high-quality pretreatment tissue was available for analysis in just 23% of the 56 ES included in our meta-analysis, representing a relatively small population. As such, the value of pIGF-1R as a predictive biomarker should be prospectively validated in future IGF-1R-related ES trials. Despite relatively small patient numbers compared to common cancers, such as breast, lung, or colon cancer, our analysis represents the largest single-institutional experience of ES patients treated with IGF-1R mAbs. It is also the first study to identify a protein biomarker of IGF-1R mAb response.

Third, for currently unexplained reasons, the response rates of our patients to single-agent IGF-1R Abs is higher than what has been reported historically by nationwide cooperative groups. One can speculate that this might be attributable to differences in patient selection, greater inclusion of adult ES patients, or perhaps reduced time between on-study baseline imaging and drug initiation, which would reduce the chance that patients’ tumors had progressed before therapy initiation. Also surprising was the fact the IGF-1R/mTORi combination reduced the percentage of patients that achieved a complete or partial response but still increased the proportion of patients with stable disease; as a result, IGF-1R/mTOR co-targeting provided superior clinical benefit (i.e., CR, PR and SD). As mTOR inhibitors are cytostatic, it is conceivable that mTORi reduced CR rates by antagonizing the cytotoxic effects of IGF-1R mAbs. However, a conclusive explanation for this unexpected finding will require prospective studies that incorporate pre- and post-treatment biopsies to more fully elucidate the antineoplastic effects of IGF-1R mAbs when using mTORi or other agents targeting the IGF-1/IGF-1R/PI3K/mTOR pathway. Of note, ES patients have infrequently been reported to respond to single-agent mTORi. Therefore, we cannot rule out that the enhanced clinical benefit for the IGF-1R/mTORi combination is secondary to mTOR inhibition rather than IGF-1R mAbs. Even if true, preclinical evidence suggests one can enhance mTORi-mediated anti-cancer effects by blocking the proximal activation of IRS-1, PI3K, or Akt.

## 4. Materials and Methods

### 4.1. Human Subjects

An IRB-approved (protocol #DR09-0245) meta-analysis was performed using data from all ES patients treated at MDACC who received an IGF-1R Ab +/- mTORi on one of the five separate clinical trials (Table 1). Patients had advanced or metastatic, histologically proven, chemo-refractory ES. Tissue specimens were evaluated using an IRB-approved laboratory protocol, and analyses were conducted in accordance with the Declaration of Helsinki. Additional details about human subjects are provided in the Appendix A.

### 4.2. Statistical Analyses

Patients’ clinical parameters were summarized with descriptive statistics and compared between single and combination therapy groups (SAS, version 9.4, SAS corporation, Cary, NC, USA). A Wilcoxon rank-sum test was used for continuous variables, and Fisher’s exact or chi-square test was used for categorical variables. The Kaplan–Meier method was utilized to obtain PFS, an OS estimate and the overall survival estimate by treatment. Cox proportional hazards regression models were utilized to assess treatment effects. Clinical benefit, as defined in our study, included patients who had a complete response, partial response, or stable disease on the study.

As 5 of the 56 patients in our meta-analysis had initially participated in a single-agent IGF-1R Ab trial before subsequently enrolling in an IGF-1R/mTORi trial when they progressed, we conducted three separate statistical analyses of PFS. The most conservative statistical approach (Method A), presented within the text of this work, assessed the response for 51 patients while excluding the 5 patients who had enrolled in both trials due to concerns that prior IGF-1R Ab exposure may have altered tumor biology and influenced the response to the IGF-1R/mTOR combination. Two alternative statistical models reached similar conclusions (see Appendix A).

### 4.3. CT & PET/CT Measurement and Interpretation

All CT and PET/CT scans were performed according to the clinical protocol at our institution. CT scans were displayed on a PACS (iSite, Philips Healthcare, Pleasanton, CA, USA). All PET/CT studies were electronically retrieved and reviewed on an MIM workstation (MIMVista, version 6.3.2, MIM Software, Cleveland, OH, USA) by an experienced board-certified radiologist blinded to the outcome. PET, CT, and fused PET/CT images were reviewed in the axial, coronal, and sagittal planes. A maximum of five tumor targets (maximum of two targets per organ) were selected. The software analysis suite (MIMVista workstation, version 6.3.2, MIM Software) that was used included a contouring PET/CT suite. Four PET-based metrics SUVmax (based on body weight), SULmax (based on lean body mass), Peak SUL, total lesion glycolysis (TLG = SUVmean × MV) were semi-automatically calculated by using auto-segmenting PET edge contouring tool of the software.

The contours were cross-checked by the visual analysis of the lesions matching with gray-scale images and sometimes adjusted for an accurate representation of the TLG of the lesions. All of the parameters were measured for the selected primary, nodal, or distant metastatic lesions. These were used for per-lesion and per-patient analysis by EORTC and PERCIST-based tumor response criteria/thresholds. Constraints like similar activity between each PET scan (± 20%), standardization against normal liver, and a similar delay between injection and acquisition (50–70 min after injection) were respected as per the PERCIST. Three dimensions of each lesion were documented on anatomic CT images for later analysis per RECIST1.1 and WHO tumor response criteria. Clinical benefit, as described in this work, refers to patients that achieved a complete or partial tumor response or stable disease using the standard definitions for WHO, RECIST, or PERCIST response criterion.

### 4.4. Immunohistochemical Staining

Formalin-fixed and paraffin-embedded tissue sections from the ES tumor tissues were deparaffinized using an alcohol gradient. After that, sections were washed and subjected to antigen retrieval for 20 min in a steamer using 1× Target Retrieval Solution (Dako, Carpentaria, CA, USA), and then allowed to cool down for 20 min to room temperature, washed, and incubated in 3% H_2_O_2_ for 15 min to block endogenous peroxidase activity. The sections were then blocked in a serum-free blocking solution (Universal LSAB^+^ kit, Dako, Carpentaria, CA, USA) for 30 min at room temperature. The primary antibodies IGF-IR (sc-713, Santa Cruz Biotechnology, Santa Cruz, CA, USA) and pIGF-IR^Y1161^ (ab39398, Abcam, Cambridge, MA, USA) diluted in blocking buffer (1:150 for IGF-IR and 1:75 for pIGF-IR^Y1161^) were added for overnight incubation at 4 °C. Subsequently, the slides were washed three times for 5 min and incubated with secondary antibody EnVision^+^ Dual Link (K406311, Dako, Carpentaria, CA, USA) for 30 min. Additional details about IHC staining are provided in the Appendix A.

## 5. Conclusions

Our meta-analysis demonstrated a significant advantage of combined cixutumumab/temsirolimus. In light of this data, and abundant evidence that single-agent IGF-1R Abs or mTORi usually invokes short-lived responses, the only viable development path today is one that capitalizes upon synergistic biologically targeted therapies [27]. While other mTORi-based combinations that jointly suppress PI3K or the IGF-1/IGF-2 ligands may be even more effective, they have not yet been tested together in this patient population. The identification of pIGF-1R as a novel predictive biomarker represents a major advance but requires confirmation by other laboratories since our sample set was small and unavailable for every patient. Critically, our independent validation of the studies by Hyun et al., (2016) has confirmed the valuable role that functional PET imaging can play as an early response biomarker of IGF-1R-directed therapies. Taken together, protein and radiologic response biomarkers provide a way to quickly predict which ES patients are most likely to benefit from IGF-1R-directed therapies. Given the rapid pace of ES tumor progression, this information delivers a precision-medicine guided approach that will help steer patients toward the therapies most likely to extend their survival.

## Figures and Tables

**Figure 1 cancers-12-01768-f001:**
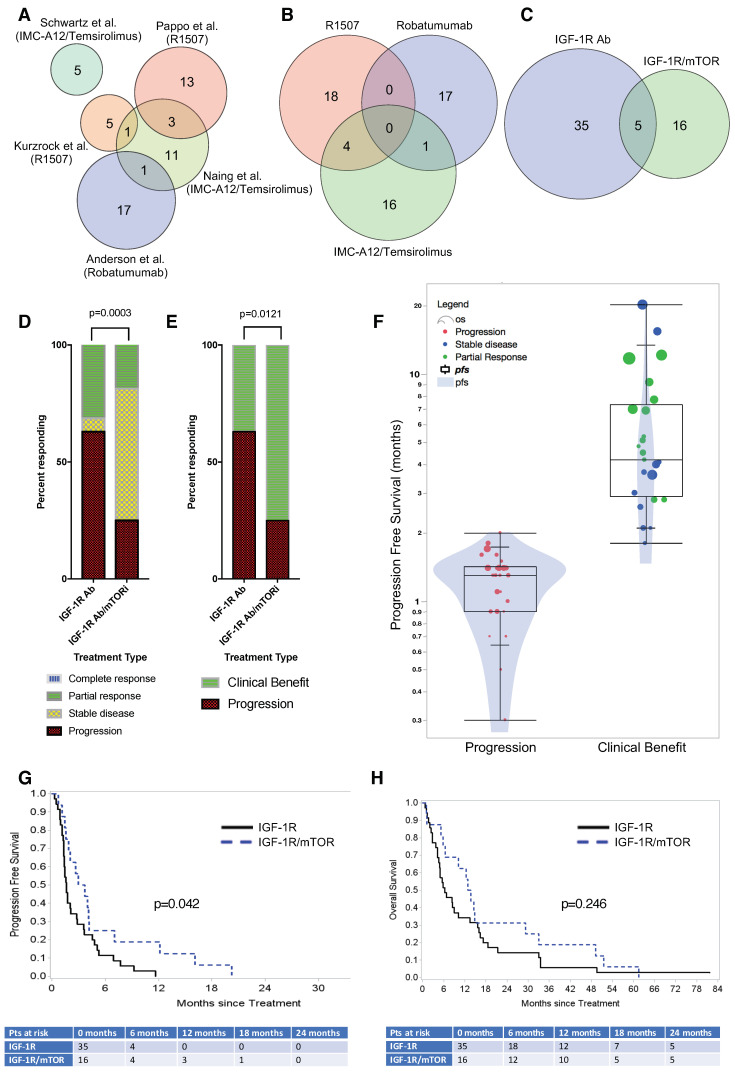
(**A**) Meta-analysis of five trials indicates that combined IGF-1R/mTORi-based therapy leads to superior clinical outcomes compared to single-agent IGF-1R mAb treatment. A Venn diagram of five IGF-1R-related clinical trials conducted at MDACC. **(B)** The two R1507 trials have been grouped for statistical analysis. (**C**) Patients treated with either R1507 or Robatumumab IGF-1R mAbs were analyzed together. Five patients received an IGF-1R/mTOR inhibitor combination after they progressed on IGF-1R Abs and were excluded from the statistical analysis. (**D**) Best clinical response by RECIST following treatment with an IGF-1R Ab vs. IGF-1R/mTOR inhibitor combination. (**E**) Similar data categorized by clinical benefit (e.g., complete or partial response and stable disease vs. progression). (**F**) Violin plot summarizing progression-free survival for all patients. Most patients progressed after 2 cycles at 6 weeks (red), whereas PFS was significantly improved in patients that achieved a partial response (green) or stable disease (blue). Each dot represents an individual patient, and the circle diameter indicates the patient’s OS duration. The superimposed box plot indicates the interquartile range. (**G**) Kaplan–Meier curves indicate a statistically significant improvement for the IGF-1R/mTOR inhibitor combination for PFS (*p* = 0.042). (**H**) The Kaplan–Meier curves of OS.

**Figure 2 cancers-12-01768-f002:**
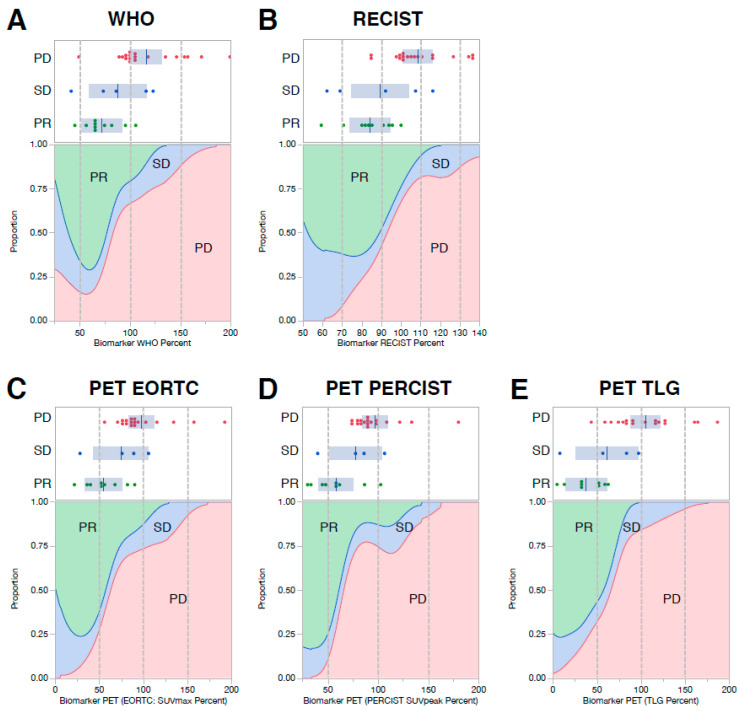
CT and PET imaging (obtained between 7–14 days following treatment with IGF-1R Abs +/− mTOR inhibitor) are predictive of each patient’s best clinical response. CT measures of response included WHO and RECIST criteria (Panels **A** and **B**, respectively). PET measures of clinical response included EORTC (Panel **C**), PERCIST (Panel **D**), and TLG (Panel **E**). Dot plots and density plots are shown for each measurement parameter. The dot plots indicate each patient’s percent response compared to the pre-treatment baseline tumor measurements, color-coded by their clinical response; progressive disease (PD; red), stable disease (SD; blue), and partial response (PR; green). The density plots beneath each dot plot show the relative probability distributions of achieving a partial response (green), stable disease (blue), or progression (red) across the entire spectrum of CT or PET responses represented by the data set. No patient achieved a complete response.

**Figure 3 cancers-12-01768-f003:**
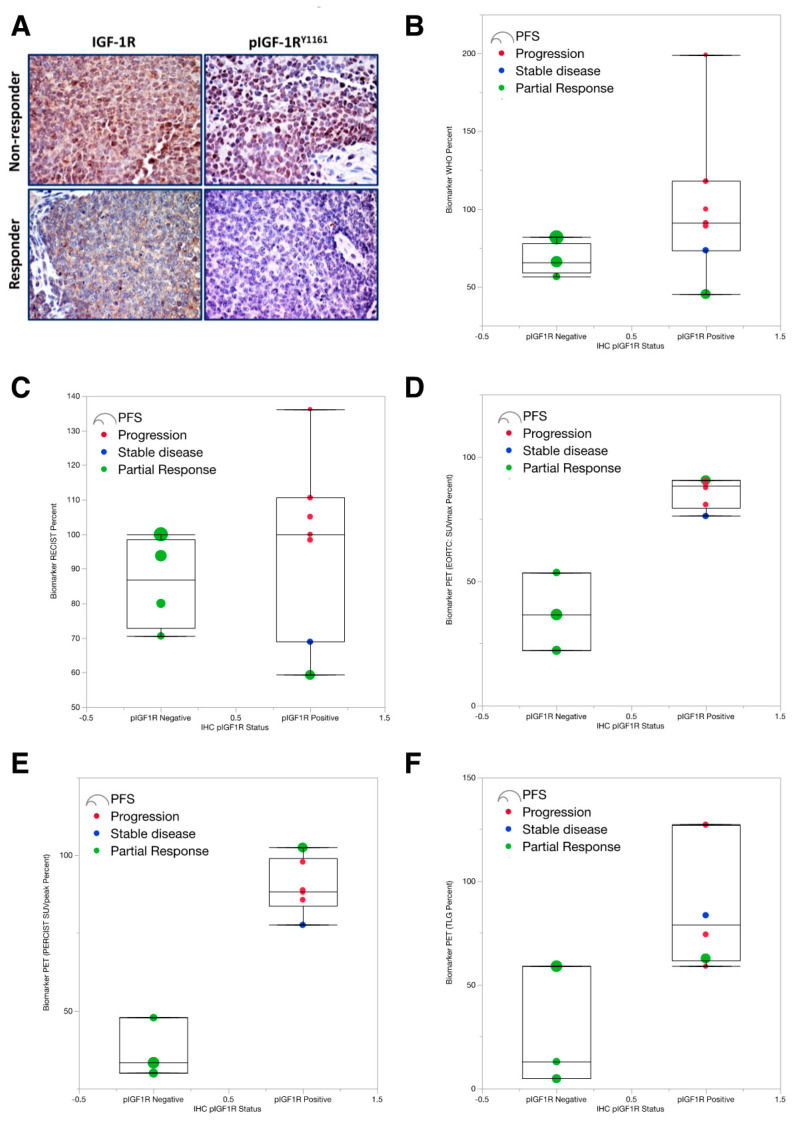
Negative pIGF-1R in pretreatment tumor specimens was associated with a better radiological response in the early PET/CTs performed within the first two-weeks of IGF-1R mAb-based therapy. (**A**) Immunocytochemical analysis (IHC) of eighteen available pre-treatment clinical specimens for phosphorylated and total IGF-1R. Post-treatment research images (performed between days 7 and 14), as assessed by the WHO response. ES specimens from two patients are shown as representative examples. The tumors from patients 1 and 2 show high levels of expression of IGF-1R (membranous and cytoplasmic). Patient 1 had significant levels of expression of pIGF-1R (membranous and cytoplasmic, as well as scattered cells with nuclear localization). This patient had poor therapeutic response. In contrast, patient 2 had almost a total lack of expression of pIGF-1R, which was associated with a superior response to therapy. Panels B–F: Radiological response stratified by p-IGF-1R status using (**B**) the WHO response, (**C**) RECIST response, **(D)** EORTC measures of PET response (**E**) PERSIST measures of PET response and (**F**) total lesion glycolysis (TLG), which is a PET volumetric parameter that reflects the total metabolic activity of the entire tumor(s). Each dot reflects an individual patient. The box plots represent a quantile. Patient response is color-coded by a partial response (green), stable disease (blue), or progression (red); no patients achieved a complete response. Circle diameter indicates the PFS duration of each patient. The EORTC response measures the sum of the maximum pixel intensity (SUV_max_) for each tumor, normalized to BSA, and expressed as a percent of baseline. PERCIST response relies upon the peak standard uptake value (SUV), normalized to lean body mass (LBM) within a 1-cm^3^ sphere of FDG-avid tumors, expressed as standard uptake lean peak (SUL_peak_); as with the EORTC criterion, SUL_peak_ values are summed and expressed as a percentage of the baseline expression value. Values < 100% indicate tumor regression, whereas values > 100% indicate tumor growth.

**Table 1 cancers-12-01768-t001:** Five IGF-1R-related clinical trials were used assessed to evaluate the clinical outcomes of ES patients treated at MDACC.

PI	IRB Number	IGF-1R Ab	mTOR Inhibitor	Sponsor	N (U.S.)	N	Age (Years)	PFS (Months)	OS (Months)	Ref.
Kurzrock	2005-0806	Teprotumumab (R1507)	N/A	Roche	9	5	23.7	2.36	9.68	[9]
Pappo	2007-0515	Teprotumumab (R1507)	N/A	Roche	115	13	31.3	2.02	9.13	[32]
Anderson	2007-0881	Robatumumab (SCH717454)	N/A	Schering-Plough	144	17	20.1	3.48	17	[33]
Naing	2007-0595	Cixutumumab (IMC-A12)	Temsirolimus	ImClone	17	11	19.7	5.94	21.5	[25]
Schwartz	2010-0191	Cixutumumab (IMC-A12)	Temsirolimus	ImClone	26	5	32.8	3.94	17.1	[28]

Shown are the nationwide study principal investigator (PI), MDACC study number, IGF-1R Ab used, mTOR inhibitor used (if applicable), demographic data, and published reference. N (U.S.) represents the number of ES patients treated nationwide, whereas N lists the number of ES patients included in the meta-analysis.

**Table 2 cancers-12-01768-t002:** Imaging-based measures of response predict tumor response and clinical benefit. Receiver operating characteristics (ROC) of CT and PET imaging are shown.

Early Imaging Response (Day 7–26)	Best Response		Clinical Benefit	
	*p*-Value	ROC	*p*-Value	ROC
WHO (CT imaging)	0.0086	0.84	0.0048	0.85
RECIST (CT imaging)	0.0104	0.85	0.0039	0.86
EORTC (PET imaging)	0.0092	0.82	0.0071	0.80
PERCIST (PET imaging)	0.0063	0.82	0.0068	0.78
TLG (PET imaging)	0.0053	0.92	0.0028	0.88

**Table 3 cancers-12-01768-t003:** Demographic and outcomes data, grouped by pre-treatment expression of pIGF-1R, as assessed by immunocytochemistry.

		IHC pIGF-1R Status		
Gender		pIGF-1R Negative	pIGF-1R Positive	All
Female		2	3	5
Male		4	9	13
All		6	12	18
Age at Diagnosis	Mean	20.8	22.6	22
	Median	22.5	21	21
	Min	10.2	12.3	10.2
	Max	31.3	46.8	46.8
Tissue Type				
Bone		5	12	17
Soft tissue		1	0	1
All		6	12	18
PFS	Mean	12.7	4.88	7.48
	Median	9.7	1.45	3.65
OS	Mean	46.2	20.2	28.9
	Median	47.3	6.5	16.8
IGF-1R best response				
Progression		0	7	7
Stable Disease		0	1	1
Partial response		5	4	9
Complete Response		1	0	1
All		6	12	18
Clinical Benefit				
Progression		0	7	7
Benefit		6	5	11
All		6	12	18
Week 6 Response (WHO)	Mean	39.3	114	88.9
	Median	25.4	120	76.9
Week 6 Response (RECIST)	Mean	66.2	93.4	84.4
	Median	63.5	105	84
Week 6 Response (SUV-Max)	Mean	62.2	108	98.6
	Median	62.2	102	74.8
Week 6 Response (SUV-Peak)	Mean	46.4	106	94.1
	Median	46.4	99.9	77.6
Biomarker (WHO Percent)	Mean	67.5	102	89.7
	Median	65.9	91	81.8
Biomarker (RECIST Percent)	Mean	86.1	96.9	93
	Median	86.9	100	98.4

Week-6 response measures indicate the tumor size at 6-weeks, expressed as a percentage of the baseline tumor parameters set at 100%. Biomarker WHO and RECIST percentages, similarly, indicate the size of the tumors obtained during the early day 9–14 PET/CTs compared the baseline tumor measurements. Values less than 100% indicate tumors that are smaller than their baseline pretreatment assessment, whereas values greater than 100% are indicative of tumor growth.

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
