# Peer review of "IGF-1R/mTOR Targeted Therapy for Ewing Sarcoma: A Meta-Analysis of Five IGF-1R-Related Trials Matched to Proteomic and Radiologic Predictive Biomarkers"

_cancers, 2020, doi:10.3390/cancers12071768_

Round 1
Reviewer 1 Report
Given the clinical dilemma that among little therapeutic options for patients with advanced ES, IGF1R targeting antibodies provided significant benefit for some patients, yet most compounds have been withdrawn from clinical development, this report is important to the field, because it underlines a perspective for combined targeting of the IGF1R-mTOR signaling cascade and suggests biological and radiological biomarkers that could help select those patients more likely to benefit.
I suggest minor revisions:
- Throughout the manuscript, the authors refer to a number of trials on IGF1R mAbs or mTORi in ES patients (by reference numbers, authors or research collaborations/institutions) and analyze patients treated on one of a subset of five trials. To better assess this, it would be helpful to provide a tabular overview of mentioned trials, beyond the subset analyzed.
- Furthermore it should be stated how the subset of 5 trials was selected (I assume they are all ES IGF1R +/-mTORi trials conducted at MDACC? However this is not stated).
- The authors state that the five different trials comprised “a similar set of patients” (line 98). It would be helpful to characterize (in brief) the study population (e.g. with Methods/Human subjects)(relapsed? progressive disease? prior therapies?).
- The sentence “measured archival tissue specimens from all available from IGF-1R mAb-treated patients” (line 106) appears awkward and should be rephrased.
- Figure 1 requires revision:
- A/B/C: Fonts are illegible
- D/E: X-axis labeling is illegible; D: what does “green” refer to, complete or partial response? Were there complete responses?
- F: what does bold italic versus standard font pfs refer to?
- G/H: what does “TX” mean? The table headers print with some kind of overlay.
- Figure 2: PR density plots are labeled yet not shown.
- The study compared cixutumumab/mTORi to different IGF1R mAbs. While this lies within the nature of studies available for analysis, this point deserves discussion.
- The authors report that contrary to expectation, “a higher proportion of patients achieved a partial response when treated using single-agent IGF-1R Abs (31.4% vs. 18.8%, p=0.0003; Fig. 1D)”. Is this true clinical effect or due to limitations of the study (such as patient number or distinct mAbs)?
- The authors demonstrate that pIGF1R IHC negative patients have better response and survival. In context with this counterintuitive result they discuss their methodology using a Y1161-directed antibody. Are there any biology-based explanatory hypothesis?
- The authors state that their findings “set the stage for drug optimization in future trials that co-target two or more proteins within the IGF-1/IGF-1R/PI3K/mTOR signaling cascade”. They analyzed by IHC pIGF1R and total IGFR expression “in addition to other proteins of the IGF1R/mTOR signaling cascade”. Which proteins were analyzed? Were additional candidate targets identified?
- The authors should double-check that all abbreviations are defined at first mention.
Reviewer 2 Report
In this paper, the authors evaluated the clinical outcome of 56 advanced-stage Ewing's sarcoma patients treated at M. D. Anderson Cancer Center from 2007-2013 on one of five different nationwide IGF-1R-based clinical trials (3 single agent IGF-1R mAbs; 2 IGF-1R/mTOR inhibitor). They found that Low pIGF-1R in the pretreatment specimens was associated with treatment response. Reduced total-lesion glycolysis more accurately predicted IGF-1R response than other previously reported radiological biomarkers. Conclusively, they stated that synergistic drug combinations, and newly identified proteomic or radiological biomarkers of IGF-1R response, may be incorporated into future IGF-1R-related trials to improve the response rate in Ewing's sarcoma patients.
This meta-analysis is very difficult to produce statistically meaningful results due to the small number of subjects. However, because it is difficult to conduct clinical research on rare diseases, it is worthwhile to establish a future research plan by analyzing existing research results. From this perspective, the value of this study can be highly appreciated.
As they mentioned, the presence or absence of pIGF-1R segregated response, but contrary to the pattern they hypothesized. The authors need to present a mechanism how low pIGF-1R in the pretreatment specimens was associated with treatment response.
I recommend that the inclusion/exclusion criteria of the five clinical studies they included are needed to add on Table 1. In clinical trials, the treatment response rates were very poor, probably because they were trials of treatment refractory patients. These statements will help readers understand.
Reviewer 3 Report
This is a well written paper describing novel biomarkers and efficacy of patients with Ewing sarcoma treated with IGF-1R/mTOR targeted therapies. The authors describe the benefits and limitations of these therapies. The results described demonstrates that this is a viable option for patients with relapse Ewing sarcoma.
Some concerns:
- Do the authors recommend early radiographic imaging for all patient treated with these therapies off of a clinical trial and does this analysis support an early consideration of switching treatment regimens for patients with relapse Ewing sarcoma?
- The mean age of patients in these studies were adults, which is to be expected with relapsed Ewing sarcoma. Do the authors hypothesize that the biology of tumors are similar and would show results to be replicated in children and adolescents as well?
Minor typos
- Line 43: elaborate pIGF-1R
- Line 329: Week 6
